## [Decision Letter · Decision Letter 0]

25 Jun 2025

PONE-D-25-27041Detecting LLM-Generated Peer ReviewsPLOS ONE

Dear Dr. Rao,

Thank you for submitting your manuscript to PLOS ONE. After careful consideration, we feel that it has merit but does not fully meet PLOS ONE’s publication criteria as it currently stands. Therefore, we invite you to submit a revised version of the manuscript that addresses the points raised during the review process.

We look forward to receiving your revised manuscript.

Kind regards,

Zheng Zhang

Academic Editor

PLOS ONE

Journal Requirements:

2. Please expand the acronym “NSF” (as indicated in your financial disclosure) so that it states the name of your funders in full.

3. Please amend your authorship list in your manuscript file to include author Vishisht Srihari Rao.

4. Please amend the manuscript submission data (via Edit Submission) to include author Vishisht Rao.

Reviewers' comments:

Reviewer's Responses to Questions

**Comments to the Author**

1. Is the manuscript technically sound, and do the data support the conclusions?

Reviewer #1: Yes

Reviewer #2: Yes

2. Has the statistical analysis been performed appropriately and rigorously? 

Reviewer #1: No

Reviewer #2: Yes

3. Have the authors made all data underlying the findings in their manuscript fully available?

Reviewer #1: Yes

Reviewer #2: Yes

4. Is the manuscript presented in an intelligible fashion and written in standard English?

Reviewer #1: Yes

Reviewer #2: Yes

5. Review Comments to the Author

Reviewer #1: Summary. To distinguish between human expert review and automatic review from an LLM, the authors propose a mechanism, invisible to humans, to detect the LLM’s reviews. For this, the authors instruct the LLM to insert a watermark. Three different watermarks are considered: Random start, Random technical term and Random citation, thus as different prompt injections.

The manuscript Is very interesting. The article's value lies in the novelty of the idea and the proposed algorithms to detect whether the review is from a human expert or an LLM. However, I doubt whether these techniques can be applied to deep learning machine training to generate a review.

Also, I have some comments about this manuscript:

1. This detection system can fail if the human reviewer chooses to make the review using an LLM and then translates the PDF output to the final document report. Have the authors considered this circumstance?

2. Additionally, if the LLM doesn’t admit instruction because it is a machine trained only to make reviews, the proposed detection system wouldn’t work

3. Concerning the Random start watermarking, I’m unsure about the statement “making the likelihood that a human-written review would coincidentally begin with the exact same phrase extremely small—at most 1 in 1,200.”. To test the truth of this sentence, I propose that the authors count different sentences, for example (“This paper studies an important problem”), across ICLR reviews or other datasets to confirm this.

4. Also, regarding the Random citation, it is not clear to me that the human reviewer will be unable to read the generated information, see if this reference is a hallucination from the LLm, and change the content.

5. The author assumes that the distribution of the watermarks follows a uniform distribution to define FPR control for a single review. Have the authors done a sample study to assess whether there is a uniform distribution?

6. What is the computational efficiency of algorithms 1,2, and 3?

7. To analyze the goodness of the proposed model, the authors experimented with OpenAI’s ChatGPT 4o, 4OpenAI’s o1-mini, Google’s Gemini 2.0 Flash, and Anthropic’s Claude 3.5 Sonnet. They chose different papers to create reviews with watermarks. In Table 3, I don’t know if the authors considered not including a watermark.

8. For a better understanding of the Control Experiment, the authors should give the number of total papers with and without watermarks

Reviewer #2: Technical soundness: The paper introduces a framework utilizing randomized watermarks, indirect prompt-injection into PDFs, and a family-wise-error-rate (FWER) test, to identify peer-review reports generated by LLMs. The statistical test is derived rigorously and is proved to control FWER without distributional assumptions on human reviews. The experimental section spans four real-world corpora (ICLR-2024 submissions, ICPRSP-2022 abstracts, PeerRead, and NSF proposals) and four frontier LLMs. Methodology and ablation studies on reviewer “defenses” are all appropriate. Limitations are acknowledged: the approach presumes reviewers paste entire PDFs into the LLM window and does not yet address partial paraphrasing across multiple prompts. Nevertheless, the data convincingly supports the paper’s claims.

Statistical analysis

Strengths

•The paper derives closed-form critical values that guarantee global α-control; proofs are complete in Appendix C.

•Simulation studies match the nominal error probabilities, demonstrating correct calibration.

Data availability

All source datasets are publicly available.

Additional comments:

One suggestion: it would be helpful to briefly discuss how the proposed technique compares to the concurrent and closely related work in arXiv:2505.16934, which also addresses LLM-generated peer reviews. Even a short comparison would clarify the distinct contributions and positioning of this work.

6. PLOS authors have the option to publish the peer review history of their article (what does this mean?). If published, this will include your full peer review and any attached files.

Reviewer #1: No

Reviewer #2: No

---

## [Author Response · Author response to Decision Letter 1]

4 Aug 2025

Reviewer Comments:

Reviewer #1:

We thank Reviewer #1 for their time and effort in reading the paper and providing feedback. Please find our responses to all their comments below.

1.

Comment: I doubt whether these techniques can be applied to deep learning machine training to generate a review.

Response: If there is a deep learning architecture specifically designed to take a paper PDF as input and produce a review as output, it may or may not contain the watermark (if the PDF with prompt injection is fed as input), depending on the architecture itself. Our goal was to design a method to reliably detect LLM-generated reviews as this is currently a very prevalent problem. We are unaware of any specific deep learning architectures that perform this task, hence, we are unable to test our pipeline on such a model.

2.

Comment: This detection system can fail if the human reviewer chooses to make the review using an LLM and then translates the PDF output to the final document report. Have the authors considered this circumstance?

Response: If the reviewer submits the PDF of the manuscript to an LLM, and asks the LLM to generate a review for the paper, the LLM will generate a review with a watermark (random start string, fake citation, technical term) embedded in the contents of the text. If this review text (i.e., the output of the LLM) is then converted to a PDF, the watermark will still remain in the text of the review, and hence can still be identified.

3.

Comment: Additionally, if the LLM doesn’t admit instruction because it is a machine trained only to make reviews, the proposed detection system wouldn’t work.

Response: If the reviewer uses a large language model designed only to produce reviews for papers and not for any other task, the generated review may or may not contain the watermark, depending on whether sufficient guard rails were set up while designing such a model. However, (a) we are not aware of any such model with strong guard rails, hence, we are unable to test our pipeline on any such model, (b) we have extensively tested our pipeline on most of the leading large language models available, and we believe that most reviewers who are not willing to put any time or effort into the reviewing process would directly use one of these.

4.

Comment: Concerning the Random start watermarking, I’m unsure about the statement “making the likelihood that a human-written review would coincidentally begin with the exact same phrase extremely small—at most 1 in 1,200.”. To test the truth of this sentence, I propose that the authors count different sentences, for example (“This paper studies an important problem”), across ICLR reviews or other datasets to confirm this.

Response: Given any human-written review with any start string, the probability with which our method will pick the same start string is at most 1/1200. This is a mathematical fact since our method picks the start string uniformly at random. We elaborate this further here (this detailed description is also present in Section 2.1.3 of the manuscript). The random start watermark is selected uniformly at random by the conference/journal organizers. The random start string has five positions as described in Table 1. Each position can take a number of different values. In our case position 1 can take 2 values, position 2 can take 5 values, position 3 can take 5 values, position 4 can take 4 values and position 5 can take 6 values. These values have been designed such that, whichever combination of values are selected at the specific positions, the random start string is plausible and coherent. The total number of random start strings is therefore 2*5*5*4*6=1200. Consider a human-written review that starts with “This paper addresses the problem”. The probability of our LLM-generated review starting with the same phrase (through prompt injection) is ½ * ⅕ * ⅕ * ¼ * ⅙ = 1/1200. Therefore, even in an extreme case where every single human-written review starts with “This paper addresses the problem”, the probability of the LLM-generated review also starting with the same phrase is 1/1200. In other words, randomness is independent of human writing styles.

5.

Comment: Also, regarding the Random citation, it is not clear to me that the human reviewer will be unable to read the generated information, see if this reference is a hallucination from the LLm, and change the content.

Response: We acknowledge that this is certainly a possibility. Our system is designed to combat the “lazy” reviewers who put minimal effort into the reviewing process. In such cases, if the reviewer does not check the generated review thoroughly, they will not identify the fake citations. We have evaluated some standard possible reviewer defenses. However, if the reviewer puts significant effort into changing and rewriting the initial LLM-generated review, it is possible that they will be able to evade detection.

6.

Comment: The author assumes that the distribution of the watermarks follows a uniform distribution to define FPR control for a single review. Have the authors done a sample study to assess whether there is a uniform distribution?

Response: We do not assume that the distribution of the watermarks follows a uniform distribution, rather, we design our algorithm that way. It is our algorithm that picks the watermark, and it picks it uniformly at random.

7.

Comment: What is the computational efficiency of algorithms 1,2, and 3?

Response: Algorithm 1 needs to check the number of occurrences of the chosen watermark in the review text. If the watermark position is fixed (as in the case of the random start, where the watermark is at the beginning of the review), the complexity is O(1). Otherwise, its complexity is a function of the length of the review text: Denoting the length of the review text by L, the complexity is O(L).

To analyze the remaining two algorithms, let us denote the number of reviews as |R| and number of elements in the watermark set as |W|. The size of the term-occurrence matrix X is then |R|*|W|. In practice, X is extremely sparse because the total number of watermark occurrences in all reviews is small. For example, consider the random start watermark, a maximum of one element in the row can contain a value of 1, since a given review can have only one possible start string. Let us denote the total number of watermark occurrences in all reviews as M. Therefore, under the practical sparsity, we have M << |R|*|W|.

Now let us consider algorithm 3. To see the maximum number of possible iterations of the loop (worst case), we can discard at most |R| reviews and |W| watermarks. Therefore, the number of iterations of the loop is O(|R| + |W|). In each iteration of the loop, we sum over all undiscarded elements in the term occurrence matrix. There are less than M undiscarded terms. Therefore, the complexity of each iteration is O(M). The complexity of algorithm 3 is therefore O(M*(|R|+|W|)).

Algorithm 2 first builds the term occurrence matrix. To do this, we first sort the list of watermark strings W in the term occurrence matrix lexicographically, which can be done in O(|W|log(|W|)). Then, for each review, we can perform a binary search for every substring in the review in the sorted W. The binary search operation for each substring in each review is O(log(|W|)). Therefore, the complexity of building the term occurrence matrix is O(|R|*log(|W|)*L) + O(|W|log(|W|)). However, similar to the example provided earlier, if the watermark position is fixed at the beginning of the review, the complexity of building the term occurrence matrix would be O(|R|*log(|W|)). Algorithm 2 then calls algorithm 3 to solve the optimization problem, which is to choose the smallest set of reviews and/or watermarks to be discarded such that the highest possible power is attained. Therefore, the complexity of algorithm 2 is O(|R|*log(|W|)*L) + O(|W|log(|W|)) + O(M*(|R|+|W|)).

8.

Comment: To analyze the goodness of the proposed model, the authors experimented with OpenAI’s ChatGPT 4o, 4OpenAI’s o1-mini, Google’s Gemini 2.0 Flash, and Anthropic’s Claude 3.5 Sonnet. They chose different papers to create reviews with watermarks. In Table 3, I don’t know if the authors considered not including a watermark.

Response: Table 3 evaluates the ability of the mentioned state of the art large language models to include the watermark in the review. In addition, we do run a control experiment where we use ICLR reviews which do not have any watermark included. In Section 3.5, we describe this control experiment, where we consider 10,000+ ICLR 2021 reviews and 28,000+ ICLR 2024 reviews. Here, there is no prompt injected into the PDF of the papers to include any watermark. Through this control experiment, we evaluate how often reviews would be falsely flagged as LLM-generated.

9.

Comment: For a better understanding of the Control Experiment, the authors should give the number of total papers with and without watermarks.

Response: For the control experiment, we consider 10,022 reviews from ICLR 2021 and 28,028 reviews from ICLR 2024. These reviews are all reviews from papers that do not have any hidden prompts, and therefore do not include any watermarks. We augment each of these sets with 100 LLM-generated reviews (these do contain a watermark, generated via papers with hidden prompts injected). This detailed description is present in Section 3.5 of the manuscript.

Reviewer #2:

We thank Reviewer #2 for their feedback and insightful suggestions. Please find our responses to their comments below.

1.

Comment: It would be helpful to briefly discuss how the proposed technique compares to the concurrent and closely related work in arXiv:2505.16934, which also addresses LLM-generated peer reviews. Even a short comparison would clarify the distinct contributions and positioning of this work.

Response: While we have added a short comparison to this paper by Yepeng Liu, Xuandong Zhao, Christopher Kruegel, Dawn Song, and Yuheng Bu in the manuscript based on the reviewer's request, we would like to note the timeline that our manuscript was submitted to arXiv on March 20, our manuscript was submitted to PLOS ONE on May 19, and the reviewer's referenced paper appeared on arXiv only after that on May 22.

---

## [Decision Letter · Decision Letter 1]

22 Aug 2025

Detecting LLM-Generated Peer Reviews

PONE-D-25-27041R1

Dear Dr. Vishisht Srihari Rao,

We’re pleased to inform you that your manuscript has been judged scientifically suitable for publication and will be formally accepted for publication once it meets all outstanding technical requirements.

Kind regards,

Zheng Zhang

Academic Editor

PLOS ONE

Additional Editor Comments (optional):

The revisions have substantially improved the quality of the manuscript. The topic is highly novel and provides profound insights into peer review in the era of artificial intelligence. I recommend the manuscript for acceptance.

Reviewers' comments:

Reviewer's Responses to Questions

**Comments to the Author**

1. If the authors have adequately addressed your comments raised in a previous round of review and you feel that this manuscript is now acceptable for publication, you may indicate that here to bypass the “Comments to the Author” section, enter your conflict of interest statement in the “Confidential to Editor” section, and submit your "Accept" recommendation.

Reviewer #1: All comments have been addressed

Reviewer #2: All comments have been addressed

2. Is the manuscript technically sound, and do the data support the conclusions?

Reviewer #1: Yes

Reviewer #2: Yes

3. Has the statistical analysis been performed appropriately and rigorously? 

Reviewer #1: Yes

Reviewer #2: Yes

4. Have the authors made all data underlying the findings in their manuscript fully available?

Reviewer #1: Yes

Reviewer #2: Yes

5. Is the manuscript presented in an intelligible fashion and written in standard English?

Reviewer #1: Yes

Reviewer #2: Yes

6. Review Comments to the Author

Reviewer #1: The authors have improved the manuscript and satisfactorily answered my questions, including some of the comments throughout the document.

Reviewer #2: (No Response)

7. PLOS authors have the option to publish the peer review history of their article (what does this mean?). If published, this will include your full peer review and any attached files.

Reviewer #1: No

Reviewer #2: No

---

## [Editor Report · Acceptance letter]

PONE-D-25-27041R1

PLOS ONE

Dear Dr. Rao,

I'm pleased to inform you that your manuscript has been deemed suitable for publication in PLOS ONE. Congratulations! Your manuscript is now being handed over to our production team.

Kind regards,

on behalf of

Dr. Zheng Zhang

Academic Editor

PLOS ONE